# Does Anxiety Affect Survival in Patients with Coronary Heart Disease?

**DOI:** 10.3390/jcm12062098

**Published:** 2023-03-07

**Authors:** Hans-Christian Deter, Wolfgang Albert, Cora Weber, Melanie Merswolken, Kristina Orth-Gomér, Christoph Herrmann-Lingen, Anna-Sophia Grün

**Affiliations:** 1Medical Clinic, Division of Psychosomatic Medicine, Charité Universitätsmedizin, Campus Benjamin Franklin, 12203 Berlin, Germany; 2German Center for Cardiovascular Research, Partner Site Berlin, 10785 Berlin, Germany; 3Psychosomatics, Medical Heart Center of Charite and German Heart Institute Berlin, Institute of Cardiothoracic and Vascular Surgery, 13353 Berlin, Germany; 4Department of Clinical Neuroscience, Karolinska Institute, 171 77 Stockholm, Sweden; 5Department of Psychosomatic Medicine and Psychotherapy, University of Göttingen Medical Center, 37075 Gottingen, Germany; 6German Center for Cardiovascular Research, Partner Site Göttingen, 37075 Göttingen, Germany

**Keywords:** anxiety, coronary heart disease, psychological intervention, 12-year follow-up

## Abstract

Introduction: Behavioral and physiological risk factors worsen the prognosis of coronary heart disease (CHD). Anxiety is known to be a psychological predictor of CHD. In this study, we investigated whether this factor is associated with all-cause mortality in CHD patients in the long term. Methods: We studied 180 patients (mean age 60.6 SD 9.2 years, 26% women) with CHD from the Berlin Anxiety Trial (BAT) and the Stepwise Psychotherapy Intervention for Reducing Risk in Coronary Artery Disease (SPIRR-CAD) study. Their cardiac and psychological risk profile was represented by standardized procedures, including the Hospital Anxiety and Depression Scale (HADS) questionnaire. Mortality outcomes were assessed using a community-based registry. Results: Of 180 patients, we obtained information on all-cause mortality in 175 (96.7%) after a mean follow-up of 12.2 years (range 10.4–16.6 years). Of all participants, 54.4% had prior myocardial infarction, 95.3% had percutaneous transluminal coronary angioplasty and 22.2% had prior coronary artery bypass graft. Most of the patients (98.4%) had New York Heart Association class I and II, 25.6% had diabetes and 38.2% were smokers. Patients had a mean HADS anxiety score of 9.7 SD 4.1 at study entrance. We found the highest HADS anxiety quartile all-cause mortality in 14%, 30.2% in the middle quartiles and 58.7% in the lowest quartile (chi^2^ 20.8, *p* = 0.001). Related to psychological mechanisms, a low level of anxiety, seemed to be a significant predictor of all-cause mortality. We found no advantage for patients who had received psychosocial therapy in terms of survival. Conclusion: These first data confirmed our hypothesis about the association of psychological risk factors with the long-term outcome of CAD patients. Future studies will clarify whether the severity of disease, age or a particular type of coping or denial mechanism are associated with the presented outcome in low-anxious patients.

## 1. Introduction

Symptoms of anxiety are frequent among patients with coronary heart disease (CHD) [1], with prevalence rates of up to 50% for panic disorder [2] and 24% for generalized anxiety disorder [3]. Symptoms of anxiety result in increased subjective suffering and decreased functioning, as well as in increased health care consumption and health care costs. They can be interpreted as maladaptive coping processes used by patients facing a situation of real threat to their lives [2]. Within a 2-year follow-up, 40% of post-myocardial infarction patients had elevated symptoms of anxiety, and anxiety was associated with more than a 50% increased risk for major adverse cardiac events (MACE) [4]. Anxiety has been linked to in-hospital cardiac complications after acute myocardial infarction [5], as well as an increased risk of mortality in patients with established CHD [6]. Increased levels of anxiety are associated with higher levels of depression [7] and an impaired health-related quality of life [8]. There is evidence for anxiety as a prognostic risk factor in CHD [9] and growing evidence that anxiety is an independent predictor of worse outcomes in cardiac populations [10,11,12,13,14].

However, evidence for anxiety as a prognostic risk factor is still conflicting and other studies have even found significant protective effects of anxiety on CHD [1,15].

Clinical and preclinical studies have gathered evidence that stress response alterations play a major role in the pathophysiology and therapy of anxiety disorders [16]. Changes in the hypothalamic pituitary adrenocortical (HPA) system and its modulation by the cortisone-releasing hormone (CRH), adrenocorticotropic hormone (ACTH), corticosteroids and their receptors are suggested to mediate the adverse effects of anxiety disorders on CHD [17,18]. Psychosomatic pathways linking anxiety and MACE in coronary heart disease seem to be an arrhythmic mechanism [19], dysfunction of the autonomic nervous system, such as reduced baroreflex cardiac control and reduced heart rate variability [2]. Whereas several peptides are known to act synergistically with CRH, the only peptide candidates in humans that inhibits the HPA system at all regulatory levels of the system seem to be the natriuretic peptides. Atrial natriuretic peptide (ANP) has been shown to inhibit the stimulated release of CRH and ACTH in vitro and in vivo. ANP receptors and immunoreactivity have been found to be critically involved in the modulation of anxiety-related behavior. Pro-ANP concentrations are correlated with low anxiety in patients with heart failure [20], and baseline NT-pro BNP indicated a stable negative association with anxiety over two years in CHD [21].

Anxiety is dependent on the patient’s physical condition, various endocrinological and autonomic stress mechanisms, as well as neuropeptide levels and the type and timing of measurement. A certain level of anxiety can be interpreted as natural and necessary for patients living with heart disease, and in severe disease, repressive coping can influence the level of anxiety [4,22].

In a review, Celano et al. [23] stated that anxiety is associated with an increased risk of mortality in patients with CAD; however, this relationship is not as strong as that of depression and may be explained partly by other clinical factors. Since anxiety has been associated with increased mortality in some studies of CHD patients [9] and with lower mortality in others [1], we wanted to investigate in patients of three psychosocial treatment studies [24,25,26], whether and in which way anxiety is related to all-cause mortality in CHD patients, up to 10 years after an acute coronary event. Additionally, we hypothesized that all-cause mortality is lower in patients who had received a psychosocial therapeutic intervention as compared to usual care.

## 2. Materials and Methods

### 2.1. Study Sample

For the study sample presented here, we collected data on survival of *n* = 180 patients who were included in one of three studies from 2004 to 2010. These were, the Berlin Anxiety trial (BAT), the Stepwise Psychotherapy Intervention for Reducing Risk in Coronary Artery Disease (SPIRR-CAD) trial (Centre Berlin) and the SPIRR-CAD pilot study. The flow chart (Figure 1) shows which patients were eligible for these three studies and the reasons for exclusion. Patients were eligible for the trial if they had documented CAD with recent coronary angiograms, with or without myocardial infarction (MI), or unstable angina pectoris (UAP), and were screened for anxiety and depression with the Hospital Anxiety and Depression Scale (HADS). Exclusion criteria were inability to speak German, severe heart failure (New York Heart Association (NYHA) class IV) or scheduled cardiac surgery within the next 3 months, severe depressive episodes or other severe or life-threatening physical or mental illness.

The pilot study consecutively included *n* = 59 CAD patients, who were in cardiologic inpatient treatment in the Charité Berlin and the German Heart Centre Berlin between December 2004 and January 2005. The natural course of anxiety and depression (HADS) in the study sample was examined over a period of 1.5 years, and group therapy as an intervention was offered and the effects of symptom reduction were tested.

The BAT was a randomized controlled trial, which included 62 patients with CHD and elevated levels of anxiety (>7 on the HADS), also recruited from Charité Berlin and the German Heart Centre Berlin, between May and December 2008. Inclusion criteria were a maximum age of 75 years, and a history of myocardial infarction (MI) or angiographically documented CHD. For the first time, this study investigated the effects of a psychotherapeutic intervention in anxious CAD patients over a course of six months [25].

The SPIRR-CAD trial was a large randomized controlled multicenter trial, comparing usual care in CAD patients to usual care, combined with a psychotherapeutic intervention. It included a total of *n* = 570 patients aged 18 to 75, with recent coronary angiograms and a depression score of >7 on the HADS [26]. N = 61 patients were recruited in Berlin between November 2008 and December 2010 in the current analyses.

More detailed descriptions of these studies can be found elsewhere [24,25,26].

### 2.2. Data Collection

Data on survival were collected using two sources of information. One source was the patient recording system of the Charité Berlin. The second source was the German civil register, where it is possible to obtain information on whether persons are deceased or still registered as alive.

Data for statistical analyses were taken from the original trials, as mentioned above.

### 2.3. Study Design

The study presented here is based on data gathered during the above-mentioned studies and a subsequent follow-up for all-cause mortality. Working hypotheses of the main follow-up examination were that psychological factors, especially depression, as well as social factors and gender, play a role in the long-term prognosis of CAD patients [27].

Due to our study aim, to demonstrate whether high anxiety is important for all-cause mortality in CHD patients 12 years after an acute coronary event, in this sub-study, we focused on the anxiety dimension.

### 2.4. Psychosocial Measures

Operationalization of our study intent was directly dependent on the data presented in the three baseline trials, which all used a very similar cardiovascular and psychological data set. Measures of all three trials were examined with the intent not to lose too much information on the one hand, and to avoid a high missing rates on the other hand. The three basic trials all used the HADS questionnaire, so we decided to use this measure to assess psychological strain. As an additional measure of social distress, we used the information regarding whether the patients lived alone or with a partner.

Anxiety was assessed by the German version of the Hospital Anxiety and Depression Scale (HADS) [28,29]. It is a two-scale instrument, with seven items for anxiety and seven items for depression. To specify cases of high, moderate, and no clinically important anxiety symptomatology, we used the HADS anxiety score to select quartiles in our sample: first quartile (<8) is assessed as no anxiety; the second and third quartile (8–12) can be evaluated as moderate anxiety and the fourth quartile (13 and higher) to assess severe anxiety [1]. Due to the sample selection in the BAT study, this scale point is one point higher compared to the original paper [29].

To expand our anxiety measurement, we used additionally:The 14-item type D scale (DS-14) [30], with the social inhibition and negative affectivity subscales.Structured rating interviews for the assessment of Axis I mental disorders, according to DSM-IV, applied by using the SCID [31] in the SPIRR-CAD subgroup and the DIPS [32] in the anxiety trial subgroup, respectively, to diagnose anxiety disorders in our sample. In this way, anxiety disorders (panic disorder, phobias, generalized anxiety disorder) were assessed in this study for the two subgroups. In the third subgroup (pilot study), a diagnostic interview could not be applied.

In the original studies, we measured HADS anxiety scores at baseline (T0) and 6 months (T2) later in the control and intervention groups. Patients in the intervention group had received 15 sessions of group-based psychotherapy and/or three individual supportive-expressive sessions of individual psychotherapy, respectively. An outcome criterion in the original studies was the reduction in anxiety scores (HADS score), which had decreased significantly in both the IG and CG groups [25,26].

### 2.5. Statistical Analyses

Baseline characteristics were described and tested using cross tables for nominal data. Metric data were tested with *t*-tests or, in case of violation of requirements of *t*-tests and not normally distributed data, with the Mann–Whitney U test, respectively. Hypotheses were tested using chi-square tests between anxiety groups and survival rates. A stepwise logistic regression and the cox regression models of proportional hazards were used to control for the effects of disease severity and the effects of medical diseases other than anxiety severity in CHD. A predefined subset of variables was entered into the analyses. Age, history of MI, LVEF and smoking were used as predictors of survival besides anxiety in patients with CHD. In addition, we included physical activity and dietary habits in the model. Missing structure was tested via Little’s test. Analyses were done using SPSS version 27. All tests were two-tailed and statistical significance was defined as *p* < 0.05 for all analyses.

The protocol of this study was approved by the Ethics Committee of the Charité, University Medicine Berlin (EA4/168/19).

## 3. Results

We studied 180 patients (mean age 60.6 SD 9.2 years, 26% women) with CHD from the three previous CHD studies in Berlin. After a mean follow-up of 12.2 years (range 10.4–16.6 years), we reached 175 (97.2%) of these patients and found that 116 (66.3%) were alive and 59 (33.7%) deceased.

### 3.1. Baseline Characteristics (Table 1)

Patient characteristics at baseline are described in Table 1 by anxiety group.

The anxiety groups did not differ in any of the demographic variables, but the patients without anxiety were sicker, had a higher NYHA class, were more likely to be smokers, had a lower BMI and a higher age compared to the patients with moderate and high anxiety.

Comparing high anxious, moderate anxious and non-anxious patients regarding somatic and psychometric variables, we found significant differences in age (Kruskal–Wallis t (180) = 16.7; *p* = 0.001); non-anxious subjects being significantly older (m = 65 years) than moderate-anxious (m = 60.3 years; *p* < 0.001) and high-anxious patients (m = 60.8; *p* < 0.001); in smoking status (Χ^2^ = 9.4; *p* = 0.009), having more smokers in the non-anxious group; and tNYHA class (Χ^2^ = 23.2; *p* = 0.001), with more patients with high NYHA in the non-anxious group.

Mean HADS anxiety scores differed between the groups, along with mean HADS depression scores. The second evaluation of HADS anxiety, 6 months after baseline evaluation, showed that the treatment and control groups had lower values compared to the first evaluation, for both. Negative affectivity on the DS 14 was found to be 88.6% in the high-anxiety group and 86.2% in the moderate-anxiety group. In the anxiety-free group, 53.2% of patients were positive for negative affectivity and 36.2% for social inhibition. In half of the patients in the no-anxiety group we also found psychological problems, which could not be detected by the HADS anxiety scale. The comparison of anxiety disorders in both groups demonstrated a high proportion of anxiety disorders in the highest quartile (52.2%), followed by the medium quartiles (29.6%). In the no-anxiety group, we found no diagnoses of anxiety disorder (Table 2). In summary, the patients of all three groups clinically had coronary disease. Most parameters were equally distributed among the three anxiety groups; however, low-anxiety patients tended to be older, sicker and had some more cardiovascular risk factors. They also seemed less psychologically burdened.

### 3.2. Six-Month Follow-Up

At the 6-month follow-up in the anxiety trial [25], significant reductions (intervention group (IG): −2.0 ± 2.3; control group (CG): −1.8 ± 2.8; *p* < 0.01) were found in both groups in the HADS anxiety scale, but no significant differences between the groups were observed. Similar reductions were seen in the SPIRR-CAD trial, Berlin Center [26]. In the pilot study [24], self-selected patients in the IG showed a higher decrease compared to self-selected CG. Adjustment for baseline differences and disease severity did not change these results. Thus, we took the baseline anxiety values in the four quartiles and looked at their predictive effects on survival after 12 years (Table 3).

### 3.3. Twelve-Year Follow-Up

Until follow-up, 59 of the 175 patients with valid data had died: 14% in the highest HADS anxiety quartile, 30.2% in the two middle quartiles and 58.7% in the lowest quartile (chi^2^ 20.8, *p* = 0.001), indicating that higher anxiety was associated with lower mortality.

To test whether disease severity had an effect on survival or interacted with the effects of anxiety, we included a predefined subset of variables in the analyses. We found a significant correlation (Spearman) of anxiety with BMI (r = 0.20), systolic blood pressure (r = −0.18) and age (r = −0.33), but not with family history of MI and other cardiovascular risk factors.

In a logistic regression analysis, our results indicated that of the included parameters (anxiety, age, previous MI, LVEF, smoking), only age (T 3.986, *p* < 0.001) and baseline anxiety (T −3.432, *p* < 0.001) had a significant influence on survival (cor. R^2^ = 0.234, F 14.6, *p* < 0.001). LVEF (*p*= 0.059) and smoking (*p* = 0.079) became nearly significant in that model. When physical activity and dietary habits (hyperlipidemia, BMI) were included in the model, again, age, anxiety and additionally LVEF (T −2.723, *p* = 0.008) became significant, but physical activity (T 1.742, *p* = 0.089), dietary habits and previous MI did not.

Additionally, a trend towards a beneficial effect of the summarized intervention of all sub-studies towards all-cause mortality could not be observed: 47 (69.1%) of the intervention patients and 68 (67.3%) of the controls survived. The change in HADS anxiety scores within 6 months of the original treatment trials in each group had no effect on survival.

## 4. Discussion

There is an ongoing discussion regarding whether anxiety or depression is an important risk factor for CHD prognosis. In this study we wanted to focus on anxiety, which is more in question as a cardiovascular risk factor than one for depression [1,7,8].

The present study shows different disease severity, cardiovascular risk factors and age between high-, moderate- and no-anxiety CHD patients. Related to psychological mechanisms, a significant predictor of all-cause mortality was a low level of anxiety. In former treatment studies of these patients, a significant decrease of anxiety scores over time in treatment and control groups has been demonstrated without advantages for the psychosocial intervention [25,26]. We wanted to evaluate whether anxiety is a predictor for all-cause mortality in this post-MI/UAP sample, as several studies have demonstrated, which included mostly cardiovascular events [1,5,6,7,8]. In their meta-analysis of 12 studies, Roest et al. [9] found that anxious patients were at risk of adverse events and new cardiac events. Anxiety was also specifically associated with all-cause mortality (OR fixed, 1.47; 95% CI, 1.02–2.13; *p* = 0.04) and cardiac mortality (OR fixed, 1.23; 95% CI, 1.03–1.47; *p* = 0.02) after two years. In contrast to these findings, we found a high survival in high- and moderate-anxious CHD patients after 12.2 years.

When examining all-cause mortality, in 97.2% of our study sample, 12 years after MI/UAP, we received no information about cardiac death, recurrent MI, or PCI in the following period, especially in the first three years.

Looking at the literature of anxiety and CHD, we should differentiate:(1)Between anxiety as risk of incident coronary heart disease, which has demonstrated convincing effects [33].(2)That after a cardiovascular event, anxiety as a trigger for new cardiovascular events in a short time perspective seems to be a risk [7,9]. An altered autonomic tone and increased susceptibility (e.g., to malignant arrhythmias), could be a possible pathway for increased mortality in patients with electrically unstable hearts after MACE.(3)In a long-term perspective, several samples have shown that anxiety is not a risk factor, 5 years after exercise testing [1,34] or 5 years after coronary stenting [15]. It is not clear to what extent self-selection for exercise testing or coronary stenting was influenced, besides by physicians’ decision, patients’ anxiety behavior or whether all patients were CHD patients or physically healthy anxiety patients [1]. In this study, we had a special selection of CHD patients who were interested in and participated in a psychosocial intervention study. The follow-up time was significantly longer at 12 years compared to former studies. It seems that anxiety in an early phase after MACE is associated with further MACE. At longer follow-up, anxiety seems to change its negative association with CHD. Additionally other factors could also play a role here. Besides low anxiety, we additionally found age and smoking in a regression analysis, and in a larger predictor model LVEF, as significant indicators of a severe disease and a well-known risk factor. Against our expectations, a previous MI, physical activity, and food intake could not predict all-cause mortality [34].

We think that the perception of anxiety is related to clinical, biological, and psychological factors that have an impact on the level of anxiety experienced by people with CHD [23]. In addition, the type of measurement and the time period in which it is carried out are important. In our study, we included only patients who were already in a stable post-treatment phase, and who had no UAP or CABG during the preceding 3 months [24,25,26], which could additionally influence the level of anxiety.

The measurement of anxiety symptoms could have an influence on our results: we applied the HADS-A as an anxiety measurement, which collects general anxiety symptoms. This was also used in a recent treatment study [35]. Other anxiety measurements, which detect panic disturbances or agora phobia, were not used. The Spielberger state-trait anxiety inventory, Beck’s anxiety or generalized anxiety disorder questionnaires could lead to different results. However, the DSM IV expert interview rating showed important validations in our study: in the subgroup of patients with the highest anxiety quartile, nearly 50% had an anxiety disorder, and in the lowest anxiety quartile, no patient received an anxiety disorder diagnosis. There is a discussion regarding whether panic disorders are dangerous [14] in developing CHD, but we found in our sample no association between panic disorder and survival.

The coping behavior and an avoiding coping style could interact and guide to low anxiety values [25]. Frasure-Smith et al. [4] found, in a secondary analysis of the M-HART trial, beneficial effects of a nursing intervention in highly anxious patients with CHD, as compared to patients with a repressive, i.e., information-avoiding coping style. In our study, a high proportion of negative affectivity and social inhibition in the no-anxiety group could demonstrate a kind of avoidance of anxiety symptoms in this group.

Anxiety could have positive influences on patients’ illness behavior. Patients with a nonphobic type of anxiety could see their doctors more frequently [1]. Moderately anxious patients might be more motivated to control their coronary risk factors [1].

We have shown that B-type natriuretic peptides are correlated to LVEF and NYHA, and this may lead to anxiolytic effects [16,21] in the no-anxiety group, which showed higher percentages of heart failure.

Anxiety in CHD patients was associated with the cortisol-awakening response: a strong and significant increase in the cortisol-awakening response (AUCi) in CHD patients with anxiety contrasted with a low AUCi after the sleep period in CHD patients without anxiety [19,27]. In these CHD patients, cortisol levels remained high after the sleep period and failed to increase. This is possibly a sign of rigidity of the endocrinological system and a persistent stress level, which could have an impact on the long-term outcome of CHD patients. The hypothesis that anxiety and high AUCi lead to an increase in MACE in CHD patients [9] does not seem to apply to long-term all-cause mortality.

In the first pilot study, we included patients aged 30 to 84 years, and in the other studies, the age of patients was 75 and younger. Therefore, we got a higher variance, but age was negatively correlated with anxiety and could have influenced both, anxiety and survival.


*Treatment Aspects*


It is important to note that intervention effects in positive treatment studies were based on stable or increasing anxiety scores in the control group, compared to the expected decrease in the intervention group. This effect could be observed only in the pilot study [24].

We investigated the effects in a group of medically stable CHD patients, eliminating the previously observed effect of spontaneous remissions that might occur in patients in the first month after MI or CABG.

We used an RCT design in two small, but homogenous, groups of CHD patients, which had focused on HADS anxiety [25] and HADS depression [26]. Thus, we investigated the effects of a psychotherapy intervention in two subgroups of distressed patients.

The combined, here-presented study shows comparable reductions of anxiety in the intervention group, but anxiety scores in the control group decreased, as well [25,26].

In contrary to our expectations, the decrease of anxiety in these studies in the intervention, as well as in the control groups, was not associated with a higher survival rate. An effect of the psychosocial intervention on survival could not be demonstrated. This was also summarized by Blumenthal et al. [35], who found in their recent anxiety treatment study within 12 months follow-up escitalopram as effective treatment for anxiety, but exercise does not appear to be effective in treating anxiety [35,36]. Treating anxiety in CHD shortly after MACE seems to be beneficial. It remains unclear which method (escitalopram, exercise, group/coping therapy) is best, as all our treated and control subjects benefited in terms of their anxiety levels from the activities undertaken during the trials. The question of up to which point in a 12-year period the treatment of anxiety in CHD patients is beneficial is comparable to looking for a needle in a haystack, given the different sources of anxiety, and is a task for future biological, psychological and cardiological studies.


*Limitation*


Our results are somewhat surprising and contrast with some studies in the literature. There are several reasons for these results:

The paradox result could be based on our study design and the relatively naïve study hypothesis, related to a simple two-factor association: anxiety and survival. Many factors could have intervened.

We have presented a selected sample of CHD patients admitted to a university hospital and the German Heart Center Berlin of the city of Berlin, with 3.7 million inhabitants and the surrounding of Brandenburg.

Patients were asked after an acute cardiovascular event if they are interested to join in a psychosocial treatment study. Hence, the motivation to join these studies was a first-selecting criteria. Several inclusion and exclusion criteria were additionally used to select this group of 180 CHD patients from three former treatment studies. As a result, a selection bias of our sample cannot be neglected.

Twenty-seven patients in the pilot study met the anxiety criteria and 18 patients had low anxiety levels. In the BAT study, all patients met the anxiety criteria (see Figure 1), which could lead to bias. The SPIRR-CAD study included 61 patients, all of whom met the depression criteria, but 12 of whom had low anxiety scores. The follow-up time of these low-anxiety patients was, thus, shorter than that in the BAT study. Data of survival in the individual trials 11, 13 and 16 years of follow-up, as well as at a 5-year follow-up, showed the same tendency: anxious patients live longer.

We have used different instruments to demonstrate anxiety, but we have not differentiated between state and trait anxiety, and the special situation of patients confronted with a severe disease: we have not differentiated between “normal” and “pathological anxiety”, which also depends on physical symptoms and the status of a patient. Therefore, we have also accepted a reporting bias. In conclusion, this study shows the difficulty of measuring and improving psychological distress in patients with CHD. One possible explanation is that anxiety is a too-heterogenous concept and influenced by many physical and psychological factors. Additionally, a study with a less pre-selected sample of heart patients could be more informative.

Interventions have shown not to be effective, and additional outcome criteria, such as active coping [25] or type D [26], could be more useful. We know that changes in the intervention, its format or its duration and intensity could change these results. Accordingly, the EUROACTION study emphasized the importance of assessing psycho-social factors, including anxiety, in CVPR programs and the inclusion of family as support in patients’ changes in behavior [37]. Alternatively, other studies focused on the behavioral treatment of stress [38]. However, the failure of several other large-scale studies [39] to find sufficient intervention effects on anxiety does account for difficulties in treating these patients. Whether exercise with behavioral therapy [40] or with escitalopram will be a better treatment method for anxious patients, will be shown the UNWIND study in the next years [35].

It is not surprising that the short time effects of presented studies are in line with longtime survival of our examined CHD patients. We found no differences in survival between treated and control patients. Whether anxiety is only a predictor of the occurrence of coronary artery disease [41], or whether it is related to biological and psychological factors after myocardial infarction [42,43] that interact with the psychological goal of treatment to reduce anxiety symptoms [44], remains unclear and may be the subject of future research [45].

## 5. Conclusions

In contrast to previous short-term follow-up studies, after major adverse coronary event (MACE), in our long-term follow-up study, high-anxious patients with CHD did not have a higher all-cause mortality than low-anxious patients. There seems to be a differential relationship between cardiovascular prognosis after MACE and anxiety: in a short-term perspective, it is associated with new cardiovascular events and mortality; in a chronic disease setting after 12 years, other chronic disease factors are more important. In addition, our study showed that elevated anxiety levels decreased over time in two studies, but there was no association with all-cause mortality and no statistically significant difference between IG and CG in all-cause mortality. Future research on anxiety after MACE and the association with recurrent MACE is needed for both short and long terms.

## Figures and Tables

**Figure 1 jcm-12-02098-f001:**
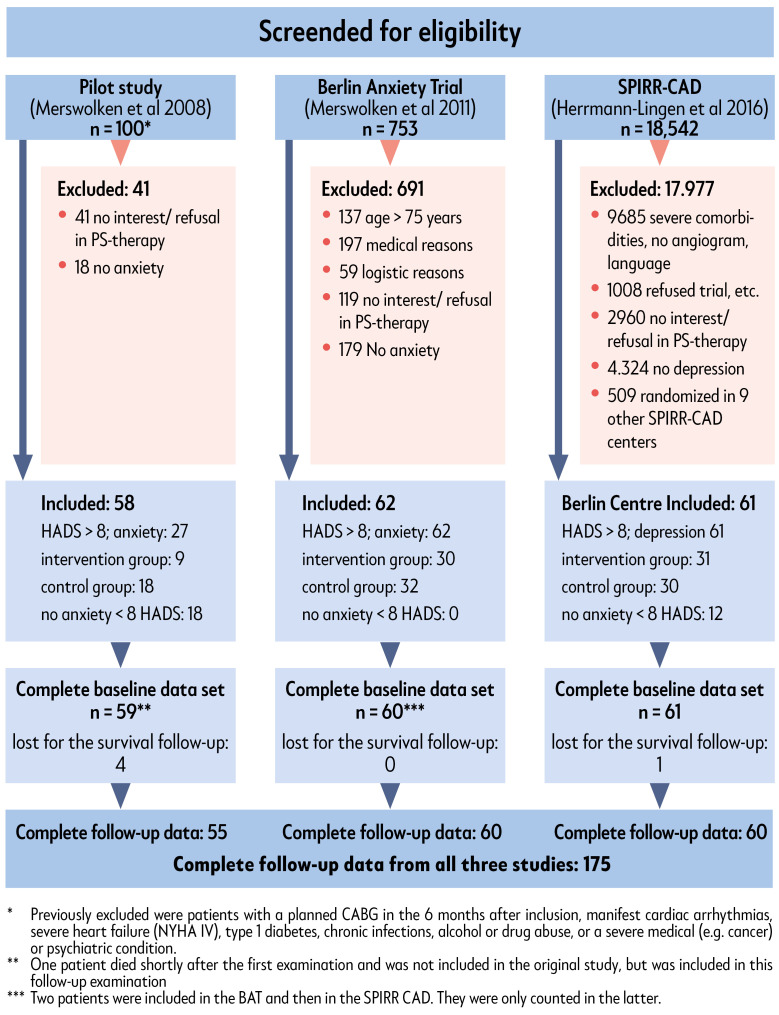
Flow chart [24,25,26].

**Table 1 jcm-12-02098-t001:** Baseline characteristics in 180 CHD patients of this study.

	No Anxiety HADS-A 0–7 N = 48		Moderate Anxiety HADS-A 8–12 N = 88		High Anxiety HADS-A 13–18 N = 44		Chi-Square *p*
	N	%	N	%	N	%	
Male sex	35	72.9	66	75	32	72.7	0.95
Married	16	55.1	56	71.8	28	70	0.47.
Socioeconomic status							0.47
Low	4	33.3	13	43.3	5	26.3	
Medium	4	33.3	13	43.3	8	42.1	
High	4	33.3	4	13.3	6	31.6	
Previous MI (*n* = 176)	20	43.5	51	58.6	23	52.3	0.21
NYHA class (*n* = 127)							<0.001
I	3	18.8	30	41.7	18	46.2	
II	10	62.5	42	58.3	21	53.8	
III + IV	3	18.8		0		0	
Hyperlipidemia (*n* = 177)	39	81.3	75	86.2	40	90.9	0.62
Hypertension (*n* = 178)	38	80.9	73	83.9	38	86.4	0.63
Diabetes mellitus (*n* = 176)	13	27.1	19	21.8	13	29.5	0.56
Smoking (*n* = 134)	14	70	27	36.0	12.	30.8	0.009
Beta-blocker (*n* = 122)	11	91.7	68	95.8	38	97.4	0.68
Statins	12	100	68	95.8	39	100	0.33
	**M**	**SD**	**M**	**SD**	**M**	**SD**	**Kruskal–Wallis** ** *p* **
Age years (*n* = 180)	65.0	9.0	60.3	8.7	60.8	9.5	<0.001 1:2; 1:3
BMI (*n* = 156)	24.2	11.9	27.4	4.1	28.5	6.5	0.23
LVEF	56.2	14.9	58.9	12.9	59.8	10.7	0.53
CCI	2.5	1.4	1.9	1.2	1.8	1.1	0.21
Aerobic exercise (min/week) (*n* = 100)	290.0	117.6	235.1	181.0	369.4	297.3	0.08

Abbreviations: BMI = body mass index (kg/m^2^), CCI = Charlson comorbidity index, M = mean, NYHA = New York Heart Association, SD = standard deviation.

**Table 2 jcm-12-02098-t002:** Anxiety Characteristics in 180 CHD patients of the BAT study.

	No Anxiety HADS A <8 *N* = 48		Moderate Anxiety HADS A 8–12 *N* = 88		High Anxiety HADS-A >12 *N* = 44		Kruskal Wallis Test *p*
	Pilot *: *N* = 36		Pilot: *N* = 17		Pilot: *N* = 4		
	AS *: *N* = 0		AS: *N* = 41		AS: *N* = 21		
	SPIRR *: *N* = 12		SPIRR: *N* = 30		SPIRR: *N* = 19		
	M	SD	M	SD	M	SD	
HADS anxiety t0 (*n* = 180)	4.5	2.2	9.8	1.3	15.0	1.6	<0.001
HADS anxiety t2 (*n* = 133) **	3.9	3.5	8.5	3.4	11.0	2.9	<0.001
HADS depression t0	5.5	3.0	8.1	3.3	11.7	3.5	<0.001
	*N*	%	*N*	%	*N*	%	**Chi square** ** *p* **
Neg. affectivity posi-tive; DS 14 (*n* = 178)	25	53.2	75	86.2	39	88.6	<0.001
Social inhibition posi-tive; DS 14 (*n* = 178)	17	36.2	60	69.0	35	79.5	<0.001
Anxiety disorder (DSM IV) interview ***; *n* = 123)							0.018
yes	0		20	29.6	21	52.2	
no	12	100	50	70.4	19	47.8	
Panic disturbance Agora phobia-(*n* = 123)							
yes	0		11	15.5	16	40.0	
no	12	100	60	84.5	24	60.0	
Phobia (*n* = 123)							
yes	0		2	1.6	1	2.5	
no	12	100	69	98.4	39	97.5	
Generalized Anxiety disorder (*n* = 123)							
yes	0		8	6.5	2	7.5	
no	12	100	63	93.5	38	92.5	

Abbreviations: DS-14 = fourteen item Type D scale, HADS = Hospital Anxiety and Depression Scale, M = mean, SD = standard deviation. * Numbers of patient in the individual sub studies: Pilot= Pilot study, AS = Anxiety Study, SPIRR = SPIRR-CAD study center Berlin. ** Activities between t0 and t2 (6 months): control condition 1 h. information or nothing, treatment: 3 h. individual treatment or 15-25 h. group intervention. *** DIPS interview in AS, SCID in SPIRR, no interview in pilot study.

**Table 3 jcm-12-02098-t003:** HADS anxiety and survival after 12 years in 175 CAD patients (chi^2^ 20.8, *p* < 0.001).

			HADS Anxiety	
		No Anxiety HADS-A: 0–7	Moderate Anxiety HADS-A: 8–12	High Anxiety HADS-A: 13–22
alive	N	19	60	37
	% within anxiety group	41.3%	69.8%	86.0%
dead	N	27	26	6
	% within anxiety group	58.7%	30.2%	14.0%
Total	N	46	86	43
	% within anxiety group	100.0%	100.0%	100.0%

## Data Availability

The data presented in this study are available on request from the corresponding author. The data are not publicly available due to the use of confidential data.

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
