# Peer review of "Does Anxiety Affect Survival in Patients with Coronary Heart Disease?"

_jcm, 2023, doi:10.3390/jcm12062098_

Round 1

Reviewer 1 Report

The authors describe an inverse relation between levels of anxiety and mortality in patients with coronary heart disease. The outcomes seem counter intuitive, as previous studies have also described higher cardiac mortality in patients with higher anxiety levels. The 

Please find my comments below:

1. Please consider shortening the introduction. The information can be summarized to make a short yet comprehensive overview of known literature and the study hypothesis.

2. Patients > 75 years of age were excluded from the study, could this affect the correlation? Although in the presented results the patients in the group without anxiety were on average older, elderly patients have been known to have higher rates of depression in other studies. 

3. It seems that the patients without anxiety were all included in the pilot study. In the SPIRR-CAD trial, only patients with HADS score >7 were included and in the Berlin Anxiety Trial patients with 'elevated levels of anxiety' were included. Could this have created a bias for the patient group without anxiety?

4. I strongly suggest the authors add a flowchart to show how patient selection from the various trials was done. 

5. Did the authors determine the short-term survival of patients in the different anxiety categories? As they suggest in the Discussion that the correlation may be opposite for the short term vs the long term this would be of additional value for the manuscript.

6. the paragraph that refers to AUCi and AUCg (line 304) in the Discussion is difficult to comprehend. Please simplify.

Author Response

Comments and Suggestions for Authors

Reviewer 1

The authors describe an inverse relation between levels of anxiety and mortality in patients with coronary heart disease. The outcomes seem counter intuitive, as previous studies have also described higher cardiac mortality in patients with higher anxiety levels.

Please find my comments below:

  1. Please consider shortening the introduction. The information can be summarized to make a short yet comprehensive overview of known literature and the study hypothesis.

The literature of CHD and anxiety is conflicting and possible mechanism for this different view on anxiety and CHD outcome should be documented. We have shortened the introduction (line 73) and summarized a short overview of known literature (line 70-74 and the aim of the study (line 77 ff..

  1. Patients > 75 years of age were excluded from the study, could this affect the correlation? Although in the presented results the patients in the group without anxiety were on average older, elderly patients have been known to have higher rates of depression in other studies. 

Patients >75 years of age were excluded in the BAT and the SPIRR-CAD study,not in the pilot study. So we got a higher variance related to age and a negative correlation with anxiety: In this study, elderly patients had lower rates of depression and anxiety

  1. It seems that the patients without anxiety were all included in the pilot study. In the SPIRR-CAD trial, only patients with HADS score >7 were included and in the Berlin Anxiety Trial patients with 'elevated levels of anxiety' were included. Could this have created a bias for the patient group without anxiety?

27 patients in the pilot study met the anxiety criteria and 18 patients had low anxiety levels. In the BAT study, all patients met the anxiety criteria (see Figure 1). The SPIRR-CAD study included 61 patients, all of whom met the depression criteria, but 12 of whom had low anxiety scores.. This  issue raised by the reviewer has now been discussed in the Discussion section (line.372-379.

  1. I strongly suggest the authors add a flowchart to show how patient selection from the various trials was done. 

A flow chart was included in figure 1

  1. Did the authors determine the short-term survival of patients in the different anxiety categories? As they suggest in the Discussion that the correlation may be opposite for the short term vs the long term this would be of additional value for the manuscript.

We have proved data of survival of the individual trials, 11, 13 and 16 years of follow-up, respectively. All showed the same tendency: anxious patients live longer in these trials. This was also the caseat five-year follow-up. (line 277) All showed the same tendency: anxious patients live longer in these studies. This was also the case at five-year follow-up. (line 275)

  1. the paragraph that refers to AUCi and AUCg (line 304) in the Discussion is difficult to comprehend. Please simplify.

We have corrected this paragraph: (line 323):

A strong and significant increase in the cortisol awakening response (AUCi) in CHD patients with anxiety contrasted with a low AUCi after the sleep period in CHD patients without anxiety [19,27]. In these CHD patients, cortisol levels remained high after the sleep period and failed to increase. Possibly this is a sign of rigidity of the endocrinological system and of a persistent stress levelThis could have an impact on the long-term outcome of CHD patients. The hypothesis that anxiety and high AUCi lead to an increase in MACE in CHD patients [9] does not seem to apply to long-term all-cause mortality.

Reviewer 2 Report

Deter et al. study found a differential relationship between cardiovascular prognosis after MACE and anxiety, association of psychological risk factors with the long-term (12 years) outcome of CAD patients.

However, this article requires several changes:

 Major comments:

1. Genetic, age and physical activity and dietary intakes are the important factors that influence both exposures and outcome. Moreover, activity and dietary intakes influence the status of anxiety; moreover, activity and dietary intakes were known to be a psychological predictor. The result base on Deter et al. study design and the relatively hypothesis related to a simple two factor association: anxiety and survival, many factors could have been intervened. It must be explained in the discussion how these factors might influence the exposures and outcome and how did you control these confounders.

2. Several inclusion and exclusion criteria were not specified in Materials and Methods, and a selection bias of our sample cannot be neglected.

3. “Patient flowchart” or “inclusion and exclusion flowchart” should be illustrated in Materials and Methods.

Minor comments:

1. Line 182, Table 1. Baseline characteristics of study. Row no. 10 “NYHA class (n=127)” and no. 17 “Smoking (n=134)” labeled chi2 attribute to statistical significance? Please illustrated chi2 for others index.

2. Line 58, MACE, such abbreviations that are unavoidable in the Introduction must be defined at their first mention there, as well as in the Conclusion.

3. Line 168, please stated “All tests were two-tailed and statistical significance was defined as p<0.05 for all analyses.”?

Author Response

Reviewer 2

Deter et al. study found a differential relationship between cardiovascular prognosis after MACE and anxiety, association of psychological risk factors with the long-term (12 years) outcome of CAD patients.

However, this article requires several changes:

 Major comments:

  1. Genetic, age and physical activity and dietary intakes are the important factors that influence both exposures and outcome. Moreover, activity and dietary intakes influence the status of anxiety; moreover, activity and dietary intakes were known to be a psychological predictor. The result base on Deter et al. study design and the relatively hypothesis related to a simple two factor association: anxiety and survival, many factors could have been intervened. It must be explained in the discussion how these factors might influence the exposures and outcome and how did you control these confounders.

The reviewer is completely right, we have analyzed these points in a new calculation (line 240 and discussed the results more clearly in the discussion section (line.288). Additionally, we have correlated physical activity, hyperlipidaemia, BMI, family history of MI and other cardiovascular risk factors with anxiety and demonstrated these in the results section (line 233)

  1. Several inclusion and exclusion criteria were not specified in Materials and Methods, and a selection bias of our sample cannot be neglected.

We have now added several inclusion and exclusion criteria in Materials and Methods (line 88-97)

  1. “Patient flowchart” or “inclusion and exclusion flowchart” should be illustrated in Materials and Methods.

A flow chart was included in figure 1 (see response to reviewer 1,4)

Minor comments:

  1. Line 182, Table 1. Baseline characteristics of study. Row no. 10 “NYHA class (n=127)” and no. 17 “Smoking (n=134)” labeled chi2 attribute to statistical significance? Please illustrated chi2 for others index.

We have changed this term.

  1. Line 58, MACE, such abbreviations that are unavoidable in the Introduction must be defined at their first mention there, as well as in the Conclusion.

We gave the information about this term in the Introduction (line 45) and Conclusion before we presented the abbreviations.

  1. Line 168, please stated “All tests were two-tailed and statistical significance was defined as p<0.05 for all analyses.”?

This sentence was included (line 171), as well as the information about additional regression parameters (line 169)